# Diagnostic Value of Chromosomal Microarray Analysis for Fetal Congenital Heart Defects with Different Cardiac Phenotypes and Extracardiac Abnormalities

**DOI:** 10.3390/diagnostics13081493

**Published:** 2023-04-20

**Authors:** Simin Zhang, Jingjing Wang, Yan Pei, Jijing Han, Xiaowei Xiong, Yani Yan, Juan Zhang, Yan Liu, Fangfei Su, Jinyu Xu, Qingqing Wu

**Affiliations:** 1Department of Ultrasound, Beijing Obstetrics and Gynecology Hospital, Capital Medical University, Beijing 100026, China; 498142914@mail.ccmu.edu.cn (S.Z.);; 2Beijing Maternal and Child Health Care Hospital, Beijing 100026, China; 3Department of Obstetric, Beijing Obstetrics and Gynecology Hospital, Capital Medical University, Beijing 100026, China; 4Department of Obstetric, Peking University People’s Hospital, Beijing 100032, China; 5Prenatal Diagnosis Center, Beijing Obstetrics and Gynecology Hospital, Capital Medical University, Beijing 100026, China; 6Department of Ultrasound, Beijing Friendship Hospital, Capital Medical University, Beijing 100032, China; 7Department of Ultrasound, Beijing Chao-Yang Hospital, Capital Medical University, Beijing 100026, China

**Keywords:** congenital heart defects, chromosomal microarray analysis, ultrasound, prenatal diagnosis

## Abstract

(1) Background: The objective of this study was to investigate the diagnostic value of chromosomal microarray analysis (CMA) for congenital heart defects (CHDs) with different cardiac phenotypes and extracardiac abnormalities (ECAs) and to explore the pathogenic genetic factors of CHDs. (2) Methods: We collected fetuses diagnosed with CHDs by echocardiography at our hospital from January 2012 to December 2021. We analyzed the CMA results of 427 fetuses with CHDs. We then categorized the CHD into different groups according to two dimensions: different cardiac phenotypes and whether it was combined with ECAs. The correlation between the numerical chromosomal abnormalities (NCAs) and copy number variations (CNVs) with CHDs was analyzed. Statistical analyses, including Chi-square tests and t-tests, were performed on the data using IBM SPSS and GraphPad Prism. (3) Results: In general, CHDs with ECAs increased the detection rate for CA, especially the conotruncal defects. CHD combined with the thoracic and abdominal walls and skeletal, thymic and multiple ECAs, were more likely to exhibit CA. Among the CHD phenotypes, VSD and AVSD were associated with NCA, while DORV may be associated with NCA. The cardiac phenotypes associated with pCNVs were IAA (type A and B), RAA, TAPVC, CoA and TOF. In addition, IAA, B, RAA, PS, CoA and TOF were also associated with 22q11.2DS. The length distribution of the CNV was not significantly different between each CHD phenotype. We detected twelve CNV syndromes, of which six syndromes may be related to CHDs. The pregnancy outcome in this study suggests that termination of pregnancy with fetal VSD and vascular abnormality is more dependent on genetic diagnosis, whereas the outcome in other phenotypes of CHDs may be associated with other additional factors. (4) Conclusions: CMA examination for CHDs is still necessary. We should identify the existence of fetal ECAs and specific cardiac phenotypes, which are helpful for genetic counseling and prenatal diagnosis.

## 1. Introduction

Congenital heart defects (CHDs) are birth defects in which the cardiovascular system is affected by many factors during embryonic development, resulting in local structural abnormalities and an actual or potential impact on cardiac function [1]. Its incidence rate is 0.4–0.8% among infants at birth, ranking first among birth defects [2]. Although progress has been made in pediatric cardiac diagnosis and health care, CHDs remain the leading cause of perinatal and infant death [3]. CHDs manifest with a wide variety and complex etiologies, and their pathogenesis has not yet been clarified. Most scholars believe that it may be caused by genetic and environmental factors and their interactions during the embryonic stage. Exploring the etiology of CHDs presents a difficult challenge. When combined with chromosomal abnormalities, the prognosis of CHD fetuses is not satisfactory due to severe extracardiac abnormalities (ECAs) or postnatal neurological development disorders. Therefore, prenatal genetic diagnosis of CHDs is recommended.

Chromosomal microarray analysis (CMA) can simultaneously detect numerical chromosomal abnormalities (NCAs) and micro chromosomal abnormalities at the whole gene level. This technique can detect copy number variations (CNVs).

In our study, we used CMA to comprehensively evaluate the detection rate for chromosomal abnormalities in CHD fetuses. Because there is a variety of CHDs, and the relationship between CHDs and genetic factors is not exclusive, we also evaluated the potential diagnostic value of CMA for different types of CHD and explored the possible genetic pathogenic factors of CHDs. We adopted a new method to explore the distribution of chromosome variation and attempted to discover its relationship with CHDs, as far as possible, so as to contribute to the prevention and treatment of CHDs.

## 2. Materials and Methods

### 2.1. Subjects

We collected fetuses diagnosed with CHDs by echocardiography at Beijing Obstetrics and Gynecology Hospital from January 2012 to December 2021; fetuses tested by CMA were taken as the research objects. NCAs and CNVs were analyzed. A total of 427 fetuses were included in the study. The gestational age range was 15–27+6 weeks, and the maternal age range was 20–47 years. This work was approved by the Medicine Ethics Committee of Beijing Obstetrics and Gynecology Hospital, Capital Medical University, Beijing Maternal and Child Health Care Hospital (No. 2018-KY-003-03), and the patients provided informed consent.

### 2.2. Classification of Cardiac Phenotypes and ECAs

A total of 427 cases of CHD were divided into isolated CHD and non-isolated CHD according to whether they were complicated by ECAs. Major ECAs were defined as abnormalities predicted to have surgical, medical or important cosmetic implications for the newborn. The ECAs were categorized according to the affected organ system (including craniofacial, neurologic, urogenital, skeletal, digestive, thoracic mass lesion, thoracic and abdominal wall and thymic abnormality). CHDs plus soft markers were categorized as isolated CHDs in our study.

Another dimension of classification was based on cardiac phenotype. The classification of cardiac phenotype was independently verified by two doctors according to the echocardiographic report and classified according to the National Birth Defect Prevention Study (NBDPS) [4]. According to the CHD phenotype, the 427 cases of CHD were categorized into ten groups, including conotruncal defects, septal defect (VSD), left ventricular outflow tract obstruction (LVOTO), right ventricular outflow tract obstruction (RVOTO), atrioventricular septal defect (AVSD), total anomalous pulmonary venous connection (TAPVC), vascular abnormality, complex CHD, heterotaxy and other CHDs (including rhabdomyoma, hydropericardium, abnormal heart rhythm and cardiac function). All fetal samples were obtained by amniocentesis to draw amniotic fluid or cord blood or by retaining skin and muscle tissue after the termination of pregnancy. Once the fetal pathogenic CNVs were confirmed, approximately 2.0 mL of peripheral venous blood was collected from their parents.

### 2.3. Chromosomal Microarray Analysis and Data Analysis

The extracted DNA samples passed the quality control test. A Cytoscan TM 750k array chip (Thermo Fisher, Waltham, MA, USA) was used for the CMA test in strict accordance with the manufacturer’s operating instructions. The chip included 200,000 SNP probes and 550,000 non-polymorphic probes, in accordance with the American College of Medical Genetics and Genomics (ACMG) and Clinical Genome Resource (ClinGen) standards and guidelines. The CNV results were divided into five categories: pathogenic CNVs (pCNVs), likely pathogenic CNVs (likely pCNVs), variations of uncertain significance (VUS), and likely benign and benign CNVs. The internationally recognized databases for reference included the UCSC Genome Browser, OMIM, decipher, DGV, PubMed, ClinGen, etc.

### 2.4. Statistical Analysis

IBM SPSS Statistics, version 26 (IBM Corp, Armonk, NY, USA) and GraphPad Prism 9 were used for the statistical analyses. Chi-square tests were used to compare groups for categorical variables. T-tests were used to analyze the difference in fragment size of CNVs in different groups of CHDs; a *p*-value of <0.05 was defined as statistically significant in the tests.

## 3. Results

### 3.1. CMA Results

The genetic diagnosis was made in 427 cases. There were 55 cases (12.9%) of NCA, 38 cases (8.9%) of pCNVs, 86 cases (20.1%) of VUS, 233 cases (54.6%) of benign CNVs and 15 cases (3.5%) of other (including chromosomal structural abnormalities and a low proportion chimera). The results from 55 cases of NCAs and 38 cases of pCNVs are summarized in Figure 1.

### 3.2. Cardiac Phenotype and ECAs in Fetal CHDs

Of the 427 CHD fetuses, 306 (71.7%) had isolated CHDs, and 121 (28.3%) had ECAs (Table 1). The ECAs were classified into eight categories; see Table 2 for details. Skeletal (32.2%, 39/121), neurologic (28.1%, 34/121) and urogenital anomalies (25.6%, 31/121) were the most prevalent ECAs in all CHDs, and a single ECA was more prevalent than multiple ECAs (65.3%, [79/121] vs. 34.7%, [42/121]) (Table 2). Conotruncal defects were the most prevalent in isolated CHDs (34.0%, 102/306) and ECAs (28.1%, 34/121); in addition, the prevalence of VSD was also higher in ECAs (26.4%, 32/121) (Table 1).

#### 3.2.1. Correlation between CA and ECAs

ECAs were more likely to have chromosomal abnormalities (CA) than isolated CHDs (36.4% [44/121] vs. 16.0% [49/306], *p* < 0.05) (Table 1). In the conotruncal defects group, the detection rate for CA was higher in ECAs compared with that of isolated CHDs (35.3% [12/34] vs. 18.1% [25/138], *p* < 0.05). In contrast, in other CHD phenotypes, there was no significant difference between the detection rate for CA with or without ECAs (*p* > 0.05).

As shown in Table 2, the incidence of CA in thymic abnormalities was 100% (5/5), and all cases involved 22q11.2DS. The incidence of thoracic and abdominal wall abnormalities was 57.2% (4/7), with three cases being trisomy 18 and one being trisomy 13. The incidence of skeletal abnormalities was 48.7% (19/39), with fourteen cases being trisomy 18, one case being trisomy 13 and four cases being pCNVs. The incidence of two or more ECAs was 47.6% (20/42), with nine cases being trisomy 18, two cases being trisomy 13, two cases being trisomy 21, one case being trisomy 9, and six cases being pCNVs.

#### 3.2.2. Correlation between CA and Phenotype of CHDs

The 427 cases of CHD were categorized into ten groups according to the CHD phenotype (see Table 1 and Table 3 for details). The highest incidence of CA in the VSD group was 41.7% (25/60), with NCA accounting for 31.7% (19/60). In the VSD group, the detection rate for trisomy 18 was 23% (14/60), with the remaining CAs being four cases of trisomy 21, one case being trisomy 13, and six cases being pCNVs.

The second highest incidence of CA was 33.3% (13/39) in the AVSD group, with 13 cases of NCA and no pCNVs detected. Of these, five cases were trisomy 21, five cases were trisomy 18, one case was trisomy 13, one case was trisomy 45, X, and one case was triploid.

The detection rates for NCA and pCNVs were similar in the conotruncal defects group (8.0% [11/138] vs. 10.1% [14/138]). However, in the DORV subgroups, the detection rate for NCA was higher than that for pCNVs (20.0% [7/35] vs. 2.9% [1/35]), while in the other four subgroups, the detection rate for pCNVs was higher than that for NCA: IAA, B (33.3% [1/3] vs. 0% [0/3]), TOF (13.2% [9/68] vs. 4.4% [3/68]), CAT (12.5% [2/16] vs. 6.3% [1/16]) and d-TGA (6.3% [1/16] vs. 0% [0/16]).

In the vascular abnormality group, only two fetuses with PLSVC had CA, with those being all trisomy 21. In RAA, the CAs were all pCNVs, and the incidence of 22q11.2DS was the highest (75%, 3/4). Genetic testing of a vascular abnormality—which may be related to trisomy 21 and 22q11.2DS—should, therefore, not be ignored.

The detection rate for pCNV in fetuses with LVOTO was higher than that for NCA (12.7% [6/47] vs. 4.3% [2/47]). The detection rates for NCA and pCNV in the RVOTO group were the same (9.5% [4/42] vs. 9.5% [4/42]). Due to the small number of cases of CA in these two groups, it is difficult to determine the distribution pattern.

The detection rate for CA in the complex CHD group was 15.2% (5/33), and there was no significant difference between the NCA and pCNVs (6.1% [2/33] vs. 9.1% [3/33]).

In summary, the cardiac phenotypes that may be related to the occurrence of NCA included VSD, AVSD, DORV, RVOTO and PLSVC. The cardiac phenotypes associated with the detection rate for pCNVs were IAA, A (50%, 1/2), IAA, B (33.3%, 1/3), RAA (28.6%, 4/14), TAPVC (25.0%, 1/4), CoA (22.2%, 2/9) and TOF (13.2%, 9/68). It is worth mentioning that the detection rates for NCA and pCNVs in heterotaxy were both 0%, and the pCNVs in AVSD were also 0%.

### 3.3. CNVs Fragment Length in CHD

To explore the CNV distribution pattern in the ten cardiac groups made up of the 427 CHD cases, violin plots were drawn for the CNV lengths (Figure 2). The identified CNVs ranged from 0.01 Mb to 8.79 Mb in length (exclusion of cases with >10 Mb), with an average length of 1.72 ± 0.23 Mb in the deletion and an average length of 1.09 ± 0.20 Mb in duplication: and there was no difference between deletion and duplication (*p* > 0.05). The CNV lengths showed no differences among the ten groups. In the violin plots, CNVs in the conotruncal defects and vascular abnormality were also concentrated around 2.5–3.5 Mb, VSD was almost always distributed between 1–10 Mb, LVOTO and RVOTO included some occurrences of extremely large fragments, and fragments in other groups were almost always below 5 Mb.

### 3.4. Detection of CNV Syndrome

Among the 38 cases of pCNVs, 29 (29/38, 76.3%) cases were related to CNV syndromes and 9 (9/38, 23.7%) cases contained the pathogenic gene.

The proportion of 22q11.2 deletion syndrome (22q11.2DS, OMIM #192430) was 4.4% (19/427), and the proportion of 22q11.2 microduplication syndrome (OMIM # 608363) was 0.2% (1/427) from cases 14 to 33 (in Appendix A). One case of 22q11.2DS was associated with a 1.1 Mb deletion on 17p13.3 deletion syndrome. Of nineteen cases, both parents in five cases agreed to be tested to determine the source of variation, confirming those five cases of 22q11.2DS were all de novo. The detection rates for 22q11.2DS in the subgroups were IAA, B (33.3%, 1/3), RAA (21.4%, 3/14), PS (11.8%, 2/7), CoA (11.1%, 1/9) and TOF (10.3%, 7/68).

Except for 22q11.2 syndrome, a total of ten CNV syndromes were found in nine fetuses. Appendix A, shows the details for the pCNV cases. Case 1 was 1p36 deletion syndrome (OMIM #607872); case 4 was 7q11.23 deletion syndrome (Williams-Beuren Syndrome, WBS, OMIM #194050); case 10 was 16p13.11 recurrent microduplication; case 11 was 17p11.2 deletion syndrome (Smith-Magenis syndrome, SMS, OMIM #182290); case 12 was Hereditary Liability to Pressure Palsies (HNPP, OMIM #162500); case 14 was 17p13.3 deletion syndrome (Miller-Dieker lissencephaly syndrome, MDLS, OMIM #247200), case 34 was 22q13.3 deletion syndrome (Phelan-McDermid syndrome, PHMDS, #606232); case 35 was cat eye syndrome (CES, OMIM #115470); case 37 was Leri-Weill dyschondrosteosis (LWD, OMIM #127300); cases 37 and 38 both involved Xq28 duplication (MECP2 duplication, OMIM #300260).

### 3.5. Other Related Pathogenic Genes

As shown in Figure 1, other pCNVs also included 5p13deletion (del), 6q26q27del, 8p23del, 10q24 duplication (dup), 11q14q22del, 11q24del, 12q23q24dup, 21q22del and Xp22del. Those pCNVs contain many pathogenic genes, including cases 2, 3, 5–9, 13 and 36, respectively (Appendix A). Among many genes, we identified five genes, *FLI1*, *NIPBL*, *DLL1*, *PTPN11* and *TBX5,* expressed in the heart and/or involved in embryonic development in four cases. The pathogenic genes in the other five cases have not yet been proven by research to be related to heart development.

### 3.6. Outcomes

A total of 325 fetuses came from terminated pregnancies: 90 cases of termination of pregnancy were confirmed by autopsy (in Appendix A), and the remaining cases were returned to the local hospital without an autopsy. Thirty-one cases were born, and seventy-one cases had unknown outcomes. Genetic abnormalities were considered having had a diagnosis.

In cases of heterotaxy, termination of pregnancy was chosen, even without a definite diagnosis. In conotruncal defects, AVSD, LVOTO, RVOTO, TAPVC and complex CHDs, the termination of pregnancy without a definite diagnosis accounted for more than 50%. In VSD, 64% of pregnancies were terminated, 42% of which were due to CA, and 22% were due to combined ECAs. In cases of vascular abnormality, only 15% of pregnancies were terminated without a definite diagnosis (Figure 3).

## 4. Discussion

CMA has emerged as a primary diagnostic tool for the evaluation of developmental delay and structural malformations in fetuses [5]. Increasing evidence has shown that CMA improves prenatal diagnostic accuracy compared to karyotyping [6,7,8]. As the primary means of prenatal CHD detection, CMA is cheap and can also detect aneuploidy and pCNVs, thus providing important clues for the prenatal diagnosis of a CHD fetus. We, therefore, need to expand its potential diagnostic efficiency as much as possible to provide a reliable plan for a prenatal consultation.

Although the occurrence of CHDs is related to genetic factors, it is not an exclusive relationship. One CHD phenotype can have multiple genetic results, and one CNV result may be related to several CHD phenotypes. The CHD phenotype is always compound. Our study shows that the incidence of ECAs in prenatal ultrasound was 28.3%, while other studies have shown that the incidence of ECAs in fetuses ranges from 50% to 66% (including autopsy) and 12–31% in ultrasound [9,10]. The definition of ECAs in our study is almost the same as the previous literature [10,11]; i.e., it included only major extracardiac anomalies, with minor anomalies being excluded. However, this study’s scholars acknowledge that the incidence of ECAs is higher because there are always some abnormalities that cannot be fully diagnosed during the fetal period. Bensemlali M et al. believed that the highest incidence of ECAs involves neurological and neurocognitive anomalies and gastrointestinal malformations [11], while Song MS et al. considered central nervous, urogenital and skeletal abnormalities [10]. The results of our study are also similar.

The CA rate in CHDs was 21.8%, and the detection rate for NCA was 12.9%. Wang et al. found that the incidence of CA was 20.8%, and more than half of the cases were NCA: our results are consistent with theirs [12]. Additionally, we found ECAs were more likely to detect CA than isolated CHDs. Scholars have also shown that the detection rate for pCNVs in CHDs plus additional structural anomalies were significantly higher than in the isolated CHD group [12]. We found that among all CHD phenotypes, conotruncal defects combined with ECAs were more likely to detect CAs compared to isolated conotruncal defects; the difference was statistically significant, which may suggest that it increases the risk of CA when combined with ECAs. On the other hand, it may be due to the small number of cases in the other groups, which failed to reflect a difference in distribution.

We found that the CAs in thoracic and abdominal wall abnormalities were all NCAs, mainly trisomy 18. More than half of the CAs in skeletal abnormalities were trisomy 18. Almost half of the CAs in multiple ECAs were also trisomy 18. It can be inferred that trisomy 18 was most related to skeletal, thoracic and abdominal wall abnormalities and multiple abnormalities in ECAs. Wang et al. thought the incidence of CA in the CHD group with multiple ECAs was higher than that in the CHD group with single ECAs; however, there was no statistical difference. Moreover, they also concluded that the incidence of CA in CHDs with craniofacial abnormalities was significantly higher than in other types of ECA. Qiao et al. analyzed CHDs using CMA and WES. They concluded that the nervous system and skeletal and urogenital anomalies received a significantly higher detection rate for the P/LP variables [13]. Based on the different conclusions by the above scholars, our study speculates that the incidence of CA is high when a CHD is combined with skeletal abnormalities and multiple ECAs and whether the abnormalities of the nervous system and urogenital system mentioned by the above scholars are related to their original incidence. In addition, a thymic abnormality was related to 22q11.2DS. Scholars observed seven fetuses diagnosed with RAA with thymus hypoplasia/aplasia, including six fetuses with 22q11.2 DS. In our study, there were also fetuses with RAA and thymus aplasia: prenatal ultrasound should improve the observation of the thymus, especially after the detection of cardiac abnormalities.

After grouping the cardiac phenotypes in our study, we found the highest incidence of CA in the VSD group, and VSD was more related to NCA, especially trisomy 18. At the same time, it is necessary to be alert to the occurrence of pCNVs. The second highest incidence of CA was in the AVSD group, and AVSD was more relevant to NCA. In conotruncal defects, DORV was the only subgroup more likely to have NCA. In addition, NCA appears in both RVOTO and PLSVC. Wang et al. believed that AVSD had the highest incidence of CA, but no pathogenic CNVs were observed [12], which is very similar to our results. This suggests that this portion of AVSD without CA may be associated with single-nucleotide variations (SNVs). Currently, studies have shown that AVSD is particularly present in patients with mutations in the *PTPN11* and *RAF1* genes [14]. In an echocardiographic study of trisomy 18 fetuses, researchers found that the most relevant cardiac phenotype was VSD [15], but in recent studies, the incidence of CA in VSD was not the highest [12,13]. The PLSVC in our study did not have intracardiac abnormalities; however, previous studies have shown that the occurrence of PLSVC is associated with cardiac abnormalities and ECAs, and the incidence of trisomy 21 was 7%. Scholars believe that the presence of PLSVC can be regarded as a CHD in the case of any associated anomaly and not merely as an anatomical variant [16].

In our study, the detection rate for pCNVs was 8.9%, which is similar to the results of previous studies [13]. The cardiac phenotypes associated with the detection rate for pCNVs were IAA, A, IAA, B, RAA, TAPVC, CoA and TOF. It is worth mentioning that the detection rates for NCA and pCNVs in heterotaxy were both 0%. Wang et al.’s results demonstrate that heterotaxy frequently associated with CA (11.7%), especially CNVs. However, other scholars conducted a CNV-seq analysis on heterotaxy and found that the incidence of CA was 4.2% (3/72) [17]. In other studies, the incidence of CA was also mostly concentrated at 2% [18,19,20]. In our study, a few cases in heterotaxy may be the reason for the low detection rate. Meanwhile, heterotaxy is necessary for further detection and to explore new genetic factors.

In the distribution of CNVs, the number of deletions was greater than duplication, and the length of deletions was slightly greater than that of duplication. Savory K et al.’s study on CHDs also found that the length and number of deletions were greater than those of duplication [21]. The reason for this phenomenon has not yet been accurately explained [22,23,24]. The phenomenon also indicates that it is easier to identify CNV deletion fragments than duplication fragments in the case of limited sequence reads. In the CHD groups, however, there were no differences between CNV lengths. This is because the dense CNVs remained <0.5 Mb in most cases, and pathogenicity is difficult to determine using the CMA technique.

The CNV syndrome that was most related to CHDs was 22q11.2 syndrome. In our study, the detection rate for 22q11.2DS was 4.4%, and for 22q11.2 microduplication syndrome, it was 0.2%. Previous studies have shown that the detection rate for 22q11.2DS in CHDs was 2.9%–9.6% in the Chinese population [25,26,27]. 22q11.2DS, including DiGeorge syndrome (DGS) and velocardiofacial syndrome (VCFS). The core clinical 22q11.2 DS phenotype was also characterized by conotruncal defects, abnormal facies, aplasia/hypoplasia of the thymus, cleft palate and hypocalcaemia [28]. In our study, we perceived that the phenotypes related to 22q11.2DS were IAA, B (33.3%), RAA (21.4%), PS (11.8%), CoA (11.1%) and TOF (10.3%). Mlynarski et al. undertook a cohort study and concluded similarly to those of our study. However, their detection rates for 22q11.2DS in TOF and IAA, B were higher and lower than those in our study, respectively (34.5% and 10.9%) [29]. Hou et al. conducted a study and found that the detection rates for 22q11.2DS in TOF and PS were 23.1% and 16.7%, respectively [25]. Zhao et al. conducted genetic analyses on 60 cases of postnatal confirmed TOF, and the detection rate for 22q11.2DS was 8.3% (5/60) [30]. Although the detection rates for 22q11.2DS in TOF are inconsistent, there is a correlation between them. The vascular ring has also been extended to the spectrum of associated clinical features in 22q11.2DS. Of course, RAA also belongs to a type of vascular ring. In addition, it should be noted that throughout our study, when a CHD was found, an ultrasound was repeatedly checked to see whether there were any abnormalities in the diameter and shape of the aorta and pulmonary artery. At the same time, attention should also be paid to the appearance of phenotypes, such as thymic abnormalities, RAA, conotruncal defects, PS or CoA.

Except for the 22q11.2 syndrome, there are many genetic factors leading to CHDs. Our study found that the following genetic disorders were related to CHDs: 1p36 deletion syndrome, WBS, SMS and MDLS [31]. There has been limited research on some CNV syndromes, such as 16p13.11 recurrent microduplication, CES and LWD, showing they are related to CHDs. Other syndromes, such as HNPP, PHMDS and Xq28 duplication, have not been shown to be associated with cardiac development but only exhibit neurodevelopmental delay [32,33]. Recent studies identified two genes forming part of a common pathway between cardiovascular and neurological development (via the Rho GTPases pathway): *LIMK1* and *MYH11* [21]. It is worth mentioning that WBS encompasses the *LIMK1* gene. In addition, 16p13.11 recurrent microduplication encompasses the *MYH11* gene. Studies have shown that *MYH11*, as a smooth muscle MHC gene, can affect arterial duct closure [34,35].

Among other pathogenic genes in pCNV, the following genes are considered to be related to the heart and/or involved in embryonic development: *FLI1*, *NIPBL*, *DLL1*, *PTPN11* and *TBX5*. (1) Hart et al. found the expression level of *FLI1* was very high in the development of vasculature and endocardium in mice [36]. (2) Northern blot analysis showed that *NIPBL* was strongly expressed in fetal and adult hearts and other tissues [37]. *NIPBL* genes were enriched for rare variants in AVSD [38]. (3) The *DLL1* gene plays an important role in the developing nervous system and somites [39]. (4) There were also reports concluding that duplication of *PTPN11* represents an uncommon cause of Noonan syndrome [40]. (5) The mutation of the *TBX5* gene causes Holt-Oram syndrome. Several studies have shown that *TBX5* associates with other gene interactions and affects heart development [41,42,43].

Our study regrets discovering that most pregnant women choose to terminate their pregnancy, despite our research center being a well-known consultation center in China. The situation regarding CHDs is complex and critical, especially conotruncal defects, AVSD, complex CHD, severe LVOTO and RVOTO, and heterotaxy syndrome. We found that CA accounts for only a portion of the prognostic outcomes of this type of CHD, and many reasons influence pregnancy outcomes, such as economic status and postoperative management. Although the development of surgery has also been explored in the last decade, many pregnant women are unwilling to bear the high cost of surgery and managed care, creating difficulty in follow-up research. We cannot predict the outcome of genetic undiagnosed (VUS) and benign results. Nevertheless, we found that pregnancy termination in VSD and vascular abnormality is more dependent on genetic diagnosis and that pregnant women prefer to continue the pregnancy if there is no clear CA.

Genome-wide association studies (GWAS) have provided evidence suggesting that CNVs significantly contribute to increased risk for congenital heart disease in conjunction with nervous system abnormalities [44,45,46]. It is also known, however, that syndromic disease-associated features may be nonspecific or erroneously attributed to cardiac lesions. The clinical manifestations of genetic syndromes are variable, even for well-defined disorders. Summarizing the clinical characteristics and gene functions of these CNV syndromes and pathogenic genes, we found that the known clinical phenotypes are always accompanied by nervous system abnormalities. Our study also concluded that pCNVs correlated with neurologic abnormality, often with no obvious ultrasonic signs that would weaken their correlation.

In the human fetus, a large part of heart and brain development occurs in a similar critical window, and thus the presence of a genetic alteration can impact both brain and heart development [47]. Prenatal ultrasound evaluation of the nervous system can only be assessed using structural abnormality and soft-marker signs. It is difficult to identify many ultrasonic signs of neurodevelopmental retardation, some of which can only be found after birth. This is a current and widespread challenge. Genetic testing potentially unlocks this situation, providing an opportunity to study the nervous system development of CHD fetuses. Given that CHDs are a lifelong condition, it is very important to have a deep understanding of the relationship between CHDs and neurodevelopmental and neurocognitive impairment in order to evaluate the condition and guide the prognosis.

Our study not only explored the genetic factors behind CHDs but also proposed a direction for tackling the diagnostic management of CHDs. Moreover, we suggest a direction for further research in order to discover correlations, expand the diagnostic value of CMA technology and provide help for clinical consultation. CNVs still contribute significantly to the etiology of CHDs. We hope to expand the number of cases and explore more useful data in the future.

## 5. Conclusions

CHDs combined with ECAs increased the detection rate for CA, especially for conotruncal defects. We focused on CHD phenotypes and ECAs associated with NCA, pCNVs and 22q11.2DS and provided assistance with prenatal diagnosis and ultrasound examination. We explored the length distribution of CNV in CHDs to provide ideas for further detection. Of the twelve CNV syndromes detected, six CNV syndromes were relatively related to CHDs, and three CNV syndromes were investigated, with the results indicating that these syndromes were related to CHDs. We noticed that some genes found in pCNVs suggested a common pathway for the development of the heart and nervous system, providing new ideas for prenatal diagnosis, genetic counseling and discussion about etiology. The pregnancy outcome in this study suggests that a termination of pregnancy for fetal VSD and vascular abnormality is more dependent on genetic diagnosis, whereas the outcome in other phenotypes of CHDs may be associated with other additional factors.

## Figures and Tables

**Figure 1 diagnostics-13-01493-f001:**
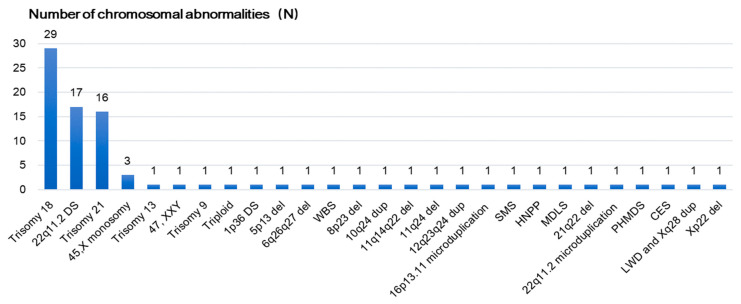
A detailed distribution map of chromosomal abnormalities. DS: deletion syndrome; del: deletion; dup: duplication; WBS: Williams-Beuren syndrome; 16p13.11 microduplication: 16p13.11 recurrent microduplication syndrome; SMS: Smith-Magenis syndrome; HNPP: Hereditary Liability to Pressure Palsies; MDLS: Miller-Dieker lissencephaly syndrome; 22q11.2 microduplication: 22q11.2 microduplication syndrome; PHMDS: Phelan-McDermid syndrome; CES: Cat eye syndrome; LWD: Leri-Weill dyschondrosteosis.

**Figure 2 diagnostics-13-01493-f002:**
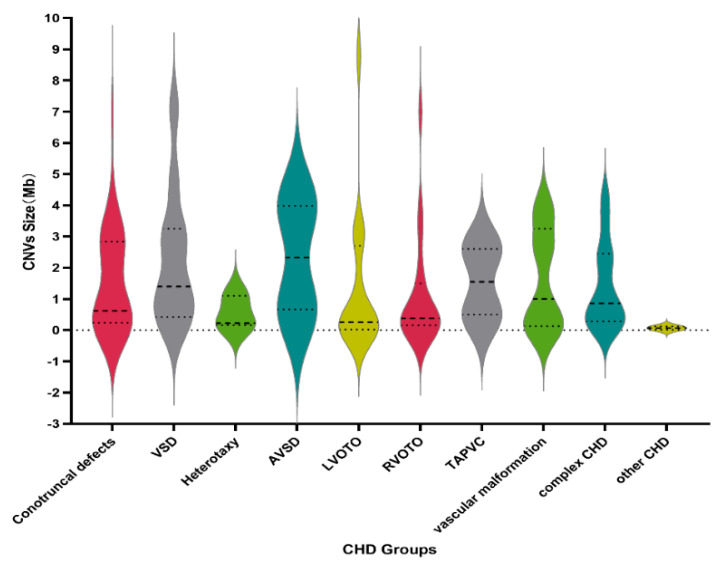
Violin plots show the distribution of the total CNV length in each group of fetal CHDs. The distribution of the lengths of all CNVs of fetuses with CHDs is represented in the figure. The thickened dotted lines indicate the median, the dotted lines represent the first and third quartiles, and the vertices of each violin plot represent outside points. VSD: ventricular septal defect; AVSD: atrioventricular septal defect; LVOTO: left ventricular outflow tract obstruction; RVOTO: right ventricular outflow tract obstruction; TAPVC: total anomalous pulmonary venous connection.

**Figure 3 diagnostics-13-01493-f003:**
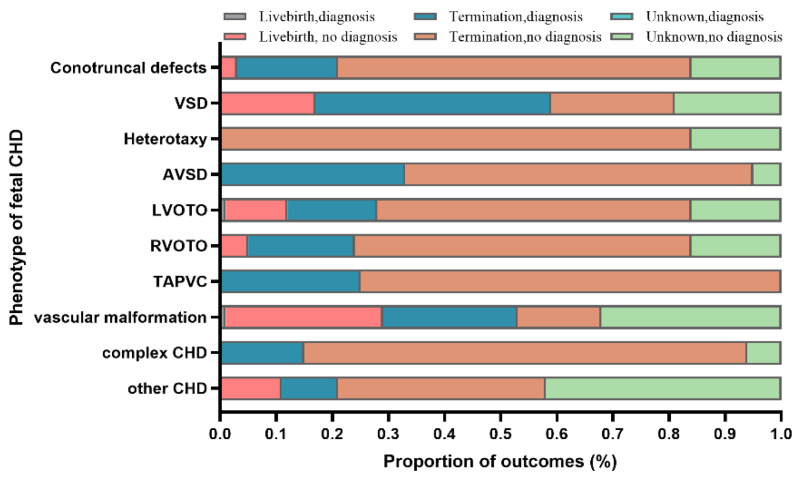
Pregnancy outcomes associated with a phenotype of fetal CHDs. A total of 138 fetuses had conotruncal defects, 60 fetuses had a septal defect, 25 fetuses had heterotaxy, 39 fetuses had AVSD, 47 fetuses had LVOTO, 42 fetuses had RVOTO, 4 fetuses had TAPVC, 22 fetuses had a vascular malformation, 33 fetuses had complex CHD, and 17 fetuses had other CHD. VSD: ventricular septal defect; AVSD: atrioventricular septal defect; LVOTO: left ventricular outflow tract obstruction; RVOTO: right ventricular outflow tract obstruction; TAPVC: total anomalous pulmonary venous connection.

**Table 1 diagnostics-13-01493-t001:** Incidence of extracardiac abnormalities (ECAs) and chromosomal abnormalities (CA) found in 427 fetuses diagnosed with congenital heart defects (CHDs).

Groups	Total		Isolated CHDs		ECAs		Total vs. Isolated vs. ECAs	Isolated vs. ECAs
	*n*	CA(*n*(%))	*n*	CA(*n*(%))	*n*	CA(*n*(%))	*p*	*p*
Conotruncal defects	138	25(18.1)	104(34.0)	13(12.5)	34(28.1)	12(35.3)	0.011	0.003
Septal defect (VSD)	60	25(41.7)	28(9.2)	11(39.3)	32(26.4)	14(43.8)	0.941	
LVOTO	47	8(17.0)	35(11.4)	5(14.3)	12(9.9)	3(25.0)	0.711	
RVOTO	42	8(19.0)	31(10.1)	4(12.9)	11(9.1)	4(36.4)	0.267	
AVSD	39	13(33.3)	23(7.5)	6(26.1)	16(13.2)	7(43.8)	0.516	
TAPVC	4	1(25.0)	3(1.0)	0(0)	1(0.8)	1(100)	0.669	
vascular abnormality	22	6(27.3)	19(6.2)	5(26.3)	3(2.5)	1(33.3)	0.990	
Heterotaxy (complex CHD)	25	0(0)	23(7.5)	0(0)	2(1.7)	0(0)		
Complex CHD (Multiple, Single ventricle)	33	5(15.2)	28(9.2)	4(14.3)	5(4.1)	1(20.0)	0.896	
Other*	17	2(11.8)	12(3.9)	1(8.3)	5(4.1)	1(20.0)	0.765	
Total	427	93(21.8)	306(100.0)	49(16.0)	121(100.0)	44(36.4)	0.000	0.000

CHDs, congenital heart defects; ECAs, extracardiac abnormalities; CA, chromosomal abnormalities; VSD, ventricular septal defect; LVOTO, left ventricular outflow tract obstruction; RVOTO, right ventricular outflow tract obstruction; AVSD, atrioventricular septal defect; TAPVC, total anomalous pulmonary venous connection. Other*, including rhabdomyoma, hydropericardium, abnormal heart rhythm and cardiac function.

**Table 2 diagnostics-13-01493-t002:** Distribution of ECAs and the incidence of CA in 121 fetuses with CHDs.

Category	*n*	CA(*n*(%))	NCA	pCNVs
Craniofacial abnormality				
Yes	24(19.8)	10(41.7)	8	2
No	97(80.2)	34(35.1)	19	15
Neurologic abnormality				
Yes	34(28.1)	10(29.4)	4	6
No	87(71.9)	34(39.1)	23	11
Skeletal abnormality				
Yes	39(32.2)	19(48.7)	15	4
No	82(67.8)	25(30.5)	12	13
Digestive abnormality				
Yes	18(14.9)	5(27.8)	4	1
No	103(85.1)	39(37.9)	23	16
Urogenital abnormality				
Yes	31(25.6)	10(32.3)	6	4
No	90(74.4)	34(37.7)	21	13
Thoracic mass lesion				
Yes	14(11.6)	6(42.9)	4	2
No	107(88.4)	38(35.5)	23	15
Thoracic & abdominal wall abnormality				
Yes	7(5.8)	4(57.2)	4	0
No	114(94.2)	40(35.1)	23	17
Thymic abnormality				
Yes	5(4.1)	5(100)	0	5
No	116(95.9)	39(33.6)	27	12
Types of ECAs				
One	79(65.3)	24(30.4)	13	11
Two or more	42(34.7)	20(47.6)	14	6
Total	121	44(36.4)	27	17

**Table 3 diagnostics-13-01493-t003:** Rate of genetic anomalies in 427 fetuses diagnosed with CHDs.

Groups and Subgroups	*n*	CA(*n*(%))	NCA(*n*(%))	pCNVs(*n*(%))	T21	T18	T13	45, X	Othe NCA	22q11.2 DS	Other CNV Syndrome	Other pCNVs
Conotruncal defects	138	25(18.7)	11(8.0)	14(10.1)	1	8	0	0	2	10	2	2
TOF	68	12(17.6)	3(4.4)	9(13.2)	0	3	0	0	0	7	1	1
CAT	16	3(18.8)	1(6.3)	2(12.5)	0	1	0	0	0	0	1	1
d-TGA	16	1(6.3)	0(0)	1(6.3)	0	0	0	0	0	1	0	0
DORV	35	8(22.9)	7(20.0)	1(2.9)	1	4	0	0	2	1	0	0
IAA, type B	3	1(33.3)	0(0)	1(33.3)	0	0	0	0	0	1	0	0
Septal defect (VSD)	60	25(41.7)	19(31.7)	6(10.0)	4	14	1	0	0	0	5	1
LVOTO	47	8(17.0)	2(4.3)	6(12.7)	0	0	0	2	0	3	1	2
Aortic stenosis	14	2(14.3)	0(0)	1(14.3)	0	0	0	0	0	1	0	1
Coarctation of aorta (CoA)	9	4(44.4)	2(22.2)	2(22.2)	0	0	0	2	0	1	1	0
HLHS	20	1(5.0)	0(0)	1(5.0)	0	0	0	0	0	0	0	1
Mitral valve dysplasia	2	0(0)	0(0)	0(0)	0	0	0	0	0	0	0	0
IAA, type A	2	1(50.0)	0(0)	1(50.0)	0	0	0	0	0	1	0	0
RVOTO	42	8(19.0)	4(9.5)	4(9.5)	3	1	0	0	0	2	0	2
HRHS	9	1(11.1)	0(0)	1(11.1)	0	0	0	0	0	0	0	1
Ebstein’s anomaly, Tricuspid valve dysplasia	8	2(25.0)	2(25.0)	0(0)	2	0	0	0	0	0	0	0
PA-IVS	8	0(0)	0(0)	0(0)	0	0	0	0	0	0	0	0
PS	17	5(29.4)	2(11.8)	3(17.6)	1	1	0	0	0	2	0	1
AVSD	39	13(33.3)	13(33.3)	0(0)	5	5	1	1	1	0	0	0
TAPVC	4	1(25.0)	0(0)	1(25.0)	0	0	0	0	0	0	1	0
vascular abnormality	22	6(27.3)	2(9.0)	4(18.2)	2	0	0	0	0	3	0	1
RAA	14	4(28.6)	0(0)	4(28.6)	0	0	0	0	0	3	0	1
Double aortic arch	2	0(0)	0(0)	0(0)	0	0	0	0	0	0	0	0
left superior vena cava	6	2(33.3)	2(33.3)	0(0)	2	0	0	0	0	0	0	0
Heterotaxy (complex CHD)	25	0(0)	0(0)	0(0)	0	0	0	0	0	0	0	0
Complex CHD (Multiple, Single ventricle)	33	5(15.2)	2(6.1)	3(9.1)	1	1	0	0	0	1	1	1
other	17	2(11.8)	2(11.8)	0(0)	0	0	0	0	0	0	0	0
Rhabdomyoma	8	0(0)	0(0)	0(0)	0	0	0	0	0	0	0	0
hydropericardium	5	2(40.0)	2(40.0)	0(0)	1	0	1	0	0	0	0	0
Abnormal heart rhythm And cardiac function	4	0(0)	0(0)	0(0)	0	0	0	0	0	0	0	0
Total	427	93(21.8)	55(12.9)	38(8.9)	17	29	3	3	3	19	10	9

CA, chromosomal abnormalities; NCA, numerical chromosomal abnormalities; pCNVs, pathogenic copy number variations; other NCA, 47,XXY, Trisomy 9 and Triploid; TOF, tetralogy of Fallot; CAT, common arterial trunk; d-TGA, d-transposition of great arteries; DORV, double outlet right ventricle; IAA, interrupted aortic arch; VSD, ventricular septal defect; LVOTO, left ventricular outflow tract obstruction; HLHS, hypoplastic left heart syndrome; RVOTO, right ventricular outflow tract obstruction; HRHS, hypoplastic right heart syndrome; PA-IVS, pulmonary atresia with intact ventricular septum; PS, pulmonary stenosis; AVSD, atrioventricular septal defect; TAPVC, total anomalous pulmonary venous connection; RAA, right aortic arch.

## Data Availability

The data supporting the findings of this study are available on request from the corresponding author.

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
