# Peer review of "Diagnostic Value of Chromosomal Microarray Analysis for Fetal Congenital Heart Defects with Different Cardiac Phenotypes and Extracardiac Abnormalities"

_diagnostics, 2023, doi:10.3390/diagnostics13081493_

Round 1

Reviewer 1 Report

An interesting, educational and insightful manuscript that has clinical merit.  However, there are editing issues that the authors should consider and address.  In the Abstract, line 6, "to two dimensions: different cardiac phenotypes and if combined with ECAs."  Line 10, "... in the conotruncal defects.  CHD combined ...".  Line 15, "... distribution of CNV was not significantly different ...".  Line 16, "... syndromes, among them, 6 syndromes ...".  In the Introduction section, paragraph 1, line 1, Congenital heart defects (CHD) is a birth defect in which the cardiovascular system is ..".  Line 8, "their interactions in the embryonic stage."  Line 11, "agnosis of CHD is recommended."  Paragraph 2, line 2, "... chromosomal micro abnormalities at the whole ...".  Paragraph 3, line 5, "...is not exclusive.  We adopt a new method ...".  In the Materials and Methods, 2.1. Subjects section, line 4, "... included in the study.  The gestational age ...".  In the 3.2. The cardiac phenotype and ECAs in fetal CHD section, line 4, "... in all CHD, and one type of ECAs was ...".  In 3.2.1. Correlation between CA and ECAs section, paragraph 1, line 4, "... p<0.05); whereas, in other CHD ...".  Paragraph 2, line 2, "... 22q11.2DS.  The incidence of thoracic ...".  Line 3, "... 1 case was trisomy 13.  The incidence of ...".  Line 5, "were pCNVs.  The incidence of two or more ...".  In 3.2.2. Correlation between CA and phenotype of CHD section, paragraph 1, lines 1 & 2, "... into ten groups (see Tables 1 & 3 for details).  The highest ...".  Paragraph 2, line 2, "cases of NCA, and ...".  Paragraph 3, line 2, "... in the subgroup of DORV, the detection ...".  Line 3, "... vs 2.9%[1/35]).  While in the other ....".  Paragraph 4, line2, "were trisomy 21.  IN the RAA, CA were ...".  Line 3, "...genetic testing of a vascular abnormality should not ...".  Paragraph 5, line 2, "... vs 4.3%[2/47]). The detection rates ...".  In the 3.3. CNVs fragment length of CHD section, paragraph 1, line 4, "... in duplication, and there were no ...".  Paragraph 2, (immediately below Figure 2), line 2, "figure.  The thickened dotted lines ...".  In the 3.4. Detection of CNV syndrome section, paragraph 2, line 6, "... CoA (11.1%, 1.9), and TOF (10.3%, 7/68)."  In 3.5. Other related pathogenic genes section, lines 2 & 3, "... 12q23q24dup, 21q22del, and Xp22del."  In the 3.6. Outcomes section, paragraph 2, lines 1 & 2, "... a definite diagnosis.  In concotruncal ...".  Line 3, '... mor than 50%.  In VSD, 64% of ...".  Lines 4 & 5, "... dur to combined ECAs.  In vascular abnormalities, only 15% ...".  Paragraph 3, line 3, "... had complex CHD, and 17 fetuses ...".  In the 4. Discussion section, paragraph 1, lines 1 & 2, '... and complex etiologies, it is always a difficult ...".  Line 4, "... for prenatal diagnosis of a CHD fetus."  Paragraphe 2, line 3, "... several CHD phenotypes.  The phenotype of CHD ...".  Line 7, "... anomalies, and minor anomalies were ...".  Lines 9 & 10, "Bensemlali M et al. believes that the ...".  Line 11, "... while Song MS et al. is considered ...".  Paragraph 3, lines 2 & 3, "... half of them were NCA, in which our study is consistent with their results [7]."  Line 6, "that in an isolated CHD group [7].  We found that among ...".  Paragraph 4, line 5, "Wang et al. think the incidence of ...".  Line 6, "...higher than that in a CHD group with single ECAs. ".  Line 8, "... types of ECAs. Qiao et al. analyzed ...".  Line 9, "... and WES.  They concluded that the nervous system, skeletal ...".  Line 14, "In addition, a thymic abnormality was ...".  Paragraph 5, line 2, "... VSD group, and the VSD was more related ...".  Line 3, "... occurrence of pCNVs.  The second highest ...".  Line 5, "...the only subgroup is more likely to have ...".  Line 6, "Wang et al. believe that AVSD has the ...".  Line 15, "... trisomy 21 was 7%. Scholars believe that ...".  Paragraph 6, line 4, "Wang et al. results demonstrated that ...".  Paragraph 8, line 2, "... rate of 22q11.2DS was 4.4%, and 22q11.2 ...".  Lines 3 & 4, "... was 2.9-9.6% in the Chinese population ...".  Line 6, "... still characterized by conotruncal defects, abnormal ...".  Line 9, "Mlynarski et al. carry on a cohort ...".,. Line 12, "Hou et al. conducted a study ...".  Line 13, "Zhao et al. conducted genetic ...".  Line 19, "... ultrasound should be repeatedly checked whether there has ...".  Paragraph 9, line 1, "... syndrome. There were many genetic ..."..,  Line 3, "... syndrome, WBS, SMS, and MDLS]24]."  Lines 3 & 4, "... have a few studies that have shown that they were ...".  Line 7, "... that WBS encompasses LIMKI gene."  Line 11, "microduplication encompasses MYH11 gene."  Paragraph 10, line 3, "TBX5 (1).  Hart et al. found the expression ...".  Paragraph 11, line 4, "... and Heterotaxy syndrome.  We found that CA accounts ...".  Paragraph 12, line 8, "... system abnormalities.  Our study also ...".  Paragraph 14, line 2, ".. for thinking of the diagnostic management of CHD."  Line 4, "... help for clinical consultation.  CNVs still make a ...".  In the 5. Conclusions section, lines 5 & 6, "... were relatively related to CHD, and three CNV syndromes have a ...".

Reviewer 2 Report

This is an interesting study on a large number of fetuses looking for CNVs and chromosome abnormalities associated to congenital heart defects.  The analysis performed seems appropiate and is reflected in the results section.  However, writing as it is makes the manuscript unintelligible and very difficult to evaluate. 

For this reason: The manuscript needs rewriting paying attention to the English language, sentences that do not make sense as they are and shortening a very lengthy discussion that is very difficult to follow and that I am not sure is necessary.

More specific:

1) There seems to be something missing at the begining of the first paragraph as it reads: "congenital heart defects (CHD), It is a birth defect that cardiovascular".

2) Verbal tenses change along the manuscript.  Please revise.

3) In section 3.3, first paragraph, I do not understand this sentence: "In the violin plots, the dense CNVs of all CHDs was still < 0.5 Mb, merited further clinical and molecular investigations."  What are dense CNVs?  And what is the meaning of this sentence?

4) In section 3.4 it states that "The five cases of 22q11.2DS were all de novo." However, there are 19 cases of 22q11.2DS and thus the sentence does not make sense.  Did they mean, 5 cases were de novo?

5) In the last sentence of section 3.5 they state: "At present, there is no research to prove that it is re-lated to cardiac abnormalities."  What is not related to cardiac abnormalities?

6) In the Discussion, I disagree with this sentence: "Some studies on CHD also found that the length and number of deletions were more than those of duplication, the reason is that both are generally believed to involve non-allelic homologous recombina-tion (NAHR)[16]." When both are expected to result from NAHR there should be equal number  of deletions and duplications.

Reviewer 3 Report

Diagnostic value of chromosomal microarray analysis for fetal congenital heart defects with different cardiac phenotypes and extra-cardiac abnormalities

The study is highly documented and provides detailed statistics over a 9-year period of congenital heart defects in the fetus. Similar studies have been carried out previously (Huang H, Wang Y, Zhang M, Lin N, An G, He D, Chen M, Chen L, Xu L. Diagnostic accuracy and value of chromosomal microarray analysis for chromosomal abnormalities in prenatal detection: A prospective clinical study. Medicine (Baltimore). 2021; 21;100(20):e25999). Cardiovascular disease mutation rate study design, tables and images, as well as associated charts are conveniently made. References must be up-dated.
